# Atomic Layer Deposition of CeO_2_ Film with a Novel Heteroleptic Ce(III) Complex

**DOI:** 10.3390/molecules29132987

**Published:** 2024-06-23

**Authors:** Wenyong Zhao, Hong Zhou, Jiahao Li, Yuchen Lu, Yuqiang Ding

**Affiliations:** International Joint Research Center for Photoresponsive Molecules and Materials, School of Chemical and Material Engineering, Jiangnan University, 1800 Lihu Road, Wuxi 214122, China; 7190610020@stu.jiangnan.edu.cn (W.Z.); 7230610020@stu.jiangnan.edu.cn (H.Z.); 6210610034@stu.jiangnan.edu.cn (J.L.); 7230610014@stu.jiangnan.edu.cn (Y.L.)

**Keywords:** heteroleptic, volatility property, ALD precursor, CeO_2_ film

## Abstract

In this paper, four heteroleptic Ce(III) complexes, including Ce(thd)_3_-phen (thd = 2,2,6,6-tetramethyl-3,5-heptanedione, phen = 1, 10—phenanthroline (**1**), Ce(thd)_3_-MEDA (MEDA = N—Methylethylenediamine (**2**), Ce(thd)_3_-MOMA (MOMA = N-(2-Methoxyethyl)methylamine (**3**), and Ce(thd)_3_-DMDE (DMDE = *N*,*N*″-dimethyl ethanol amine (**4**), were synthesized and characterized with ^1^H-NMR, elemental analysis, and X-ray single-crystal diffraction. The thermogravimetric analysis and vapor pressure results indicated that the complexing ability of a nitrogen-containing bidentate ligand with a cerium ion was stronger than that of a mixed oxygen-nitrogen-containing bidentate ligand. Complex **2** was selected as an ALD precursor to deposit a CeO_2_ film on a SiO_2_/Si (100) wafer. The self-limited deposition results demonstrated that complex **2** was a potential ALD precursor.

## 1. Introduction

Thin films of rare-earth (RE) oxide are functional materials that have attracted much attention in complementary metal-oxide-semiconductor (CMOS) and memory applications [1,2]. Among various RE oxide materials, cerium dioxide (CeO_2_) has the potential to replace conventional silicon dioxide (SiO_2_) material, because of the high dielectric constant (23–52), high refractive index (2.2–2.8), high dielectric strength (~25 Mv/cm), and moderate bandgap (3.0–3.6 eV) [3,4,5,6,7]. In addition, CeO_2_ material can also be used in applications such as solid oxide fuel cells, optical coatings, catalysts for water splitting, and air purification [8,9,10,11,12,13,14,15].

To date, various techniques have been used to deposit CeO_2_ films, such as sol-gel [16], electrochemical vapor deposition (EVD) [17], molecular beam epitaxy (MBE) [18], metal-organic chemical vapor deposition (MOCVD) [19,20,21], sputtering [22], electron beam evaporation [23], atomic layer epitaxy (ALE), and atomic layer deposition (ALD) [24,25,26]. Among these technologies, ALD was considered the most attractive technology to deposit CeO_2_ films, owing to an adjustment of the number of deposition cycles to control the thickness of the films, thickness uniformity, and composition controllability and the ability to deposit high-uniformity thin films on highly non-planar wafers. ALD has a self-limiting growth mechanism and unique surface saturation in chemical surface reactions, and the process was strongly dependent on the chemical properties of precursors [27,28,29]. Therefore, high thermal stability, excellent volatility, and a low melting point of the precursor were necessary.

In recent years, various scientific groups have explored a variety of Ce-based ALD precursors to deposit cerium oxide including *β*-diketonates [30], alkoxides [31], cyclopentadienyls [32], amidinates, and guanidinates [33,34]. Ce-cyclopentadienyl complexes and their derivatives have shown high melting points, low volatility, and poor thermal stability. Ce(^i^PrCp)_3_ (^i^PrCp = isopropylcyclopentadienyl) was explored as a cerium precursor to deposit CeO_2_ films with high carbon impurities [35]. An alkoxide-based precursor could only be applied successfully in liquid injection delivery ALD systems, as shown for Ce(mmp)_4_ (mmp = 1-methoxy-2-methyl-2propoxide) [36]. It exhibited poor volatility and thermal stability. Amidinate- and guanidinate-based cerium precursors with excellent volatility and high thermal stability have been demonstrated to be promising for ALD of CeO_2_. These include Ce(N-^i^Pr-AMD)_3_ (N-^i^Pr-AMD = N, N″-diisopropylacetamidianato), and Ce(dpamd)_3_ (dpamd= N, N′-diisopropyl-2-dimethylamido-guanidinato) [37,38]. In addition, these precursors were sensitive to water and air compared to diketonate-based precursors, which require more elaborate handling procedures. Furthermore, a heteroleptic cerium precursor was liquid at room temperature and could be evaporated at 145 °C in the case of Ce(^i^PrCp)_2_(N-^i^Pr-AMD), which was sensitive to water and air [39,40]. The most commonly applied precursors for ALD of CeO_2_ thin films were *β*-diketonate chelates of cerium due to their ease of synthesis, insensitivity to water and oxygen, and high productivity in handling. However, only Ce(thd)_4_ (thd = 2,2,6,6-tetramethyl-3,5-heptanedione) was reported as an ALD precursor to deposit films, but *β*-diketonate precursors have a high melting point and poor volatility [41,42].

Recently, heteroleptic precursors have received more attention [43]. With heteroleptic precursors, the metal center was bonded with the best properties of different types of ligands to obtain potential precursors with high thermal stability, excellent volatility, and low melting points [44,45,46]. Cerium *β*-diketonates showed better volatility and thermal and moisture stability than complexes with alkoxides, cyclopentadienyls, amidinates, and guanidinates. There are two types of cerium ligand precursors, with the cerium ions in either the 3^+^ or 4^+^ oxidation state. Ce(IV) complexes such as Ce(thd)_4_ and Ce(mmp)_4_ showed low volatility and reactivity. Ce(III) complexes such as Ce(N-^i^Pr-AMD)_3_ (177–180 °C) with high melting points were prone to oxidation, making them difficult to obtain and keep in a pure state [38]. Introducing neutral ligands helps to satisfy the requirement of a high coordination number (coordination saturation) and decreases the intermolecular interactions with neighboring hydrogen atoms [47]. In our previous study, La(thd)_3_-DMEA (DMEA = N, N′-dimethylethylenediamine) as a heteroleptic complex was synthesized and used as an ALD precursor to deposit La_2_O_3_ film. It showed that the introduced neutral ligands saturated the metal center and improved volatility and thermal stability [48]. Leskelae T, et al. synthesized Ce(thd)_3_-phen (phen = 1, 10—phenanthroline) with a high melting point (210 °C) as a heteroleptic ALE precursor, which was sublimated in two stages, indicating phen of the adduct molecules during the sublimation and the complexing ability of the oxygen *β*- diketonate ligand with a cerium ion was stronger than that of a nitrogen-containing bidentate heterocyclic neutral ligand [49]. However, the strength of the bond between nitrogen-containing and mixed oxygen-nitrogen-containing bidentate neutral ligands with cerium ions has not been thoroughly studied. Therefore, the development of heteroleptic cerium dipivaloymethanates with bidentate neutral ligands was investigated. These are promising as ALD or CVD precursors.

In this paper, four heteroleptic cerium complexes were synthesized using different types of bidentate neutral ligands and an Hthd ligand. These included Ce(thd)_3_-phen (phen = 1, 10—phenanthroline (**1**), Ce(thd)_3_-MEDA (MEDA = N-Methylethylenediamine (**2**), Ce(thd)_3_-MOMA MOMA= N-(2-Methoxyethyl)methylamine (**3**), and Ce(thd)_3_-DMDE (DMDE= N,N″-dimethyl ethanol amine (**4**). Complex **1** was characterized by ^1^H-NMR, and complexes **2**–**4** were characterized using ^1^H-NMR, elemental analysis, and X-ray single-crystal diffraction. Their properties, including melting points, thermal stability, and vapor pressure, were investigated by thermogravimetric analysis (TGA) and melting point measurements. The TGA data show that the introduction of neutral ligands could improve the volatility of heteroleptic Ce(III) *β*- diketonate complexes. Complex **2** was evaporated at 170 °C and used as a liquid precursor at that temperature to deposit CeO_x_ film on a SiO_2_/Si (100) wafer. The composition and surface morphology of the film as deposited were analyzed using various techniques. An X-ray photoelectron spectroscope (XPS) was used to analyze the composition of the films, atomic force microscopy (AFM) was used to analyze the surface morphologies of the films, grazing incidence X-ray diffraction (GIXRD) was used to analyze the crystalline phase of the films, and cross-sectional scanning electron microscopy (SEM) was used to verify the ellipsometry model and calculate the growth per cycles (GPCs).

## 2. Result and Discussion

### 2.1. Crystal Structure Descriptions

All complexes were insensitive to air and water. Complexes **1**–**4** were obtained as materials after recrystallization from an *n*-hexane solution at −20 °C, with high yields (85%, 83%, 80%, and 85%, respectively). The structures of complexes **2**–**4** were characterized by the X-ray crystal structure. The ^1^H-NMR spectra of complexes were also investigated, and the ^1^H-NMR results exhibited that the number of peaks was consistent with the construction of the target product in the solution. The ^1^H-NMR data, elemental analysis, and the single-crystal X-ray diffraction results demonstrated that complexes **2**–**4** were consistent with the target product.

Table 1 and Table 2 summarize the crystallographic data and selected bond lengths for complexes **2**–**4**, respectively. And the crystallographic data of complexes **2**-**4** was provided in Appendix A. To further verify the structure of complex **3**, single-crystal X-ray diffraction was performed at a low temperature. Table 1 and Table 2 and Figure 1a show the crystallographic data, selected bond lengths, and coordination environment of complex **2**, respectively.

As shown in Figure 1a, the single-crystal X-ray results revealed that complex **2** crystallizes in the monoclinic crystal system in the p2_1_-c space group with an eight-coordinate monomer, which contains three thd^−^ ligands and one MEDA neutral ligand. This shows that in its distorted octahedral geometry, the center of the cerium ion was coordinated with six oxygen atoms of the thd- ligands and two nitrogen atoms from the MEDA ligand to form Ce-O bonds and Ce-N bonds, respectively. The average length of Ce-O bonds from the chelate thd^−^ ligand was 2.427 Å, which was not significantly different from the 2.472 Å found in [Ce_2_(etbd)_6_(tetraglyme)] [47]. The average Ce-O (thd) and Ce-N (MEDA) bond lengths compare well with that found in the lanthanum [48] and are in good agreement with the value expected for a Ce^III^ ion. The observed bond length trend agrees well with the ionic radii variation from Ce^III^ (1.11 Å) through to La^III^ (1.15 Å) [50]. The length of the chelated ligand has a Ce-N distance of 2.753 Å, which was shorter than the chelate-bridging ligands because of lower steric interactions. The length of Ce-N bonds from the chelated ligands of the NH_2_ group was 2.722 Å, which was shorter than the lengths of Ce-N bonds (2.753 Å) that came from the chelate ligand in the NH group in complex **2**. The NH_2_ group of the chelated neutral ligand may also contribute to complex stabilization due to intramolecular hydrogen bonds.

As shown in Figure 1b, complex **3** adopts an eight-coordinate structure with a mononuclear structure that is similar to that of complex **2**, with three thd^−^ ligands and one MOMA neutral ligand. Like complex **3**, the center of the cerium ion was coordinated with six oxygen atoms from three thd^−^ ligands and an oxygen atom and a nitrogen atom from one MOMA ligand, respectively, to form Ce-O bonds and Ce-N bonds. The average bonds distances of Ce-O of thd^−^ ligands (2.422 Å) were shorter than that of the MOMA ligand (2.675 Å), indicating that the complexing ability of an oxygen *β*- diketonate ligand with a cerium ion was stronger than that of a mixed oxygen-nitrogen-containing neutral ligand. The length of the Ce-N bond in complex **3** was 2.709 Å, which was similar to the length of the bond found in complex **2**.

As seen in Figure 1c, complex **4** exhibited a mononuclear and an eight-coordinate structure with a distorted octahedral geometry around a cerium center ion, which was surrounded by six oxygen atoms of three thd- ligands and an oxygen atom and a nitrogen atom of one DMDE ligand to form Ce-O bonds and Ce-N bonds, respectively. The average length of Ce-O bonds in complex **4** was 2.413 Å, which was not significantly different from the 2.422 Å found in complex **3** or the 2.427 Å found in complex **4**. The length of the Ce-O bond from the DMDE was 2.660 Å, which was similar to the length of the bond found in complex **3**. The length of the Ce-N bond in DMDE was 2.792 Å, which was longer than that of MOMA (2.709 Å), due to the DMDE having stronger steric hindrance than the MOMA. Although numerous attempts were made, a high-quality crystal of complex **1** could not be obtained, despite its easy solubility in organic solvents such as *n*-hexane. Based on the crystal structure and ^1^H-NMR data, the structure of complex **1** appeared reasonable. Additionally, the ^1^H-NMR data of complex **1** demonstrated its structure conformed to the target product. Clearly, all the above results illustrate the rationality of the structure of complexes **1**–**4**.

### 2.2. Thermal Properties of Complexes

For the complex to be a potential ALD candidate, high thermal stability, excellent volatility, and a low melting point were required and measured by TGA and differential scanning calorimetry (DSC). The complex was in the liquid state below the melting point temperature in ALD, which was preferred due to constant surface areas and gas-phase precursor concentration.

A low melting point was used as the first factor to evaluate the extent of intermolecular interactions in complexes. The melting points of the complexes were analyzed by differential scanning calorimetry (DSC) and a melting point apparatus. The results are presented in Figure 2 and summarized in Table 3. The endothermic peaks in the DSC correspond to the melting points of complexes **1**, **2**, **3**, and **4**, which are found to be 210 °C, 165.9 °C, 144.9 °C, and 128.5 °C, respectively. It was found that those results agreed with the melting points measured by the melting point apparatus. Complex **1** has a higher melting point (210 °C) than complexes **2**–**4**. This can be explained by the introduction of small-molecule neutral ligands, which can reduce the melting point. 

As shown in Figure 2a,b, the TG results showed that complexes **1** and **2** had single-step volatilization with no decomposition, which was ideal behavior for potential ALD precursors. Residual mass from TG can help to assess the volatilization and decomposition of complexes. The residual masses of complexes **1** and **2** were 0.96% and 1.97%, respectively. The lower residual mass shows that complexes **1** and **2** can be cleanly evaporated and not decomposed. The temperature of 50% mass loss (T_50_), which was obtained from the TG curves, was a key piece data with which to evaluate the volatility of the complex. Low T_50_ values indicated excellent volatility. The T_50_ values of complexes **1** and **2** were 318 °C and 268.9 °C, respectively. This indicates that complex **2** was more volatile than complex **1**, which was ascribed to the lowest neutral ligand molecular weight among the monomeric complexes. And the complexing ability of nitrogen-containing bidentate small-molecule neutral ligand weights with a cerium ion was stronger than that of an aromatic nitrogen-containing bidentate neutral ligand. The NH_2_ group may also affect the stability of the complex due to intramolecular hydrogen bonds. The T_50_ and residual mass results show that complex **2** exhibits excellent thermal stability and volatility compared to complex **1**. Those results indicated that the introduction of small-molecule neutral ligands can increase the volatility of the complex.

In Figure 2c, the first weight loss step (12%) occurs at 182.7 °C, which could be attributed to the loss of one neutral ligand (MOMA) from complex **3**. The DSC curve shows four sharp peaks, the first and third endothermic peaks at 94.1 °C and 222 °C, respectively, which can be associated with some change in crystal structure, since no mass loss was observed. The second peak at 144.9 °C corresponds to the melting of complex **3**. As seen in Figure 2d, the TG curves of complex **4** show similar behavior to complex **3**. This weight loss difference of 0.5%, when compared to complex **3**, is most probably due to the higher molecular mass of the DMDE. The decomposition of complexes **3** and **4** was evidence that the complexing ability of an oxygen *β*- diketonate ligand with a cerium ion was stronger than that of a mixed oxygen-nitrogen-containing bidentate neutral ligand, which was not suitable as ALD precursor. 

All the results indicated that the introduction of small-molecules neutral ligands can reduce the melting point and the complexing ability of neutral ligands with a cerium ion and improve the volatility of the complex. Moreover, the complexing ability of nitrogen-containing bidentate small-molecule neutral ligand weights with a cerium ion was stronger than that of an aromatic nitrogen-containing bidentate neutral ligand. And the complexing ability of a nitrogen-containing bidentate ligand with a cerium ion was stronger than that of a mixed oxygen-nitrogen-containing bidentate ligand.

A standard definition for “volatility” of the complex was the temperature at which that complex reaches 0.1 Torr of vapor pressure. Vapor pressure-temperature plots were obtained by thermogravimetry. The theoretical basis of the TG procedure is the Langmuir and Antoine equation, for which benzoic acid was chosen as a standard. As shown in Figure 3, the temperature of complex **2** (153 °C) was lower than that of complex **1** (212 °C), indicating that complex **2** has more excellent volatility than complex **1**.

Among the newly synthesized cerium complexes, complex **2** has the lowest residue and a relatively lower T_50_ value and sublimation temperature (170 °C) under reduced pressure (0.3 Torr) as a liquid precursor at that temperature, indicating that it has sufficient volatility and thermal stability to be used as a potential ALD precursor.

### 2.3. Growth Characteristics of CeO_x_ Deposition

Based on the promising results obtained from the thermal stability and vapor pressure characterization of the complexes, the next target was to evaluate the precursors for ALD applications. As a representative case, complex **2** was selected as an ALD precursor to carry out the ALD process for CeO_x_. An ALD experiment on a SiO_2_/Si(100) wafer was performed with complex **2** using O_3_ as a co-reactant.

As shown in Figure 4a, the growth of films was carried out under a temperature range of 280–340 °C with the pulse of complex **2** (12 s) and O_3_ (12 s). When the deposition temperature was increased to 300 °C, the growth rate initially increased, and then the growth-per-cycle (GPC) was maintained at a nearly constant value of approximately 0.52 Å/cycle in the deposition temperature range of 300–320 °C. When the deposition temperature was increased to 340 °C, the GPC increased sharply to 0.67 Å/cycle. The temperature range of 300–320 °C was the ALD temperature window in this deposition process. The deposition temperature was below 300 °C to obtain the low growth rate due to the lack of adequate activation energy for the surface chemical reaction. For the high growth rate at the deposition temperature above 340 °C, the reason was attributed to the thermal decomposition of complex **2**.

The complex **2** pulse times were varied from 0 to 18 s while all other parameters were kept constant: the O_3_ pulse time at 12 s, the deposition temperature at 300 °C, and the number of deposition cycles at 300. It can be seen in Figure 4b that the 12 s pulse time for complex **2** was long enough to achieve surface saturation. The corresponding film growth rate expressed as the GPC was 0.52 Å/cycle. The O_3_ pulse time was varied from 0 to 18 s, maintaining all other parameters constant: complex **2** pulse time at 12 s, deposition temperature at 300 °C, and number of deposition cycles at 300. As shown in Figure 4c, a saturation behavior was observed at 12 s of O_3_ pulse time having a GPC of 0.52 Å/cycle. Finally, the dependency of the film thickness on the number of cycles was analyzed as seen in Figure 4d, maintaining all other parameters constant: complex **2** pulse time at 12 s, O_3_ pulse time at 12 s, and deposition temperature at 300 °C. The film thickness was linear with the number of cycles, and the GPC and the obtained fit values were approximately 0.52 Å/cycle and 0.9999, respectively. All results in terms of validating ALD growth characteristics further confirm that complex **2** was suitable as an ALD precursor.

### 2.4. Characterization of CeO_x_ Films

The surface morphology of the as-deposited film was observed by SEM and AFM. As shown in Figure 5a, the thickness of the films was 110 nm, which was measured by an ellipsometer and deposited on the SiO_2_/Si (100) wafer at 300 °C for 2200 cycles, and the result corresponded to the cross-sectional data by SEM. The results demonstrated that the thickness of the film as measured by the ellipsometer and the ellipsometry model was reasonable. In Figure 5b, the AFM results showed that the film was continuous with no cracks, and the film surface roughness had a low value (RMS = 0.479 nm), indicating that the film had a smooth surface. The composition and crystalline phase of the film deposited at 300 °C were investigated by grazing incidence X-ray diffraction (GIXRD). As shown in Figure 5c, the GIXRD pattern was obtained for a 110 nm thick film grown at 300 °C for 2200 cycles. Diffraction peaks appeared at 2θ = 28.6°, 33.6°, and 47.5°, which were attributed to the (111), (200), and (220) reflections of polycrystalline cerium oxide, respectively [51,52]. 

To further analyze the composition of the film, XPS was carried out. As the survey spectra of the film grown at 300 °C in Figure 6a illustrate, the results indicated that the major components of the film were Ce, O, and C. However, it can be seen in Figure 6b that the C content decreased to 0% after Ar^+^ sputtering for 2 min. This result indicated that the present C might be derived from the atmosphere (Table 4). Figure 6c shows the high-solution Ce3d XPS spectra of the film, and it was found that six peaks occurred at 882.07 (v), 888.20 (v′), 898.10 (v″), 900.43 (u), 907.33 (u′), and 916.36 eV (u″). The former three peaks were ascribed to Ce 3d_5/2_, whereas the latter three peaks belonged to Ce 3d_3/2_ [53]. Following the reported XPS data for CeO_x_, the six peaks were assigned to Ce^4+^ ions [54]. As seen in Figure 6d, the O 1s core levels of the as-deposited CeO_x_ films show that the peak at 531.7 eV might be ascribed to the surface chemisorbed oxygen in the form of surface -OH/-CO_3_^2−^ [53]. Another peak at 526.15 eV corresponded to Ce^4+^-O, which was consistent with previous reports for CeO_2_ [39]. In addition, Ar^+^ sputtering would change the chemical structure of the sample layer by XPS, so the XPS spectrum of the film was not analyzed after sputtering [55]. The XPS results indicated that the as-deposited films were CeO_2_, and these could be compared well with the GIXRD results. Undoubtedly, all these results demonstrate that complex **2** could be successfully used as a potential ALD precursor to deposit CeO_2_ film on SiO_2_/Si (100) wafers.

## 3. Experimental

### 3.1. Materials

N-Methylethylenediamine (MEDA), N-(2-Methoxyethyl)methylamine (MOMA), N,N″-dimethyl ethanol amine (DMDE), 1,10-phenanthroline (phen), sodium hydroxide (NaOH), 2,2,6,6-tetramethyl-3,5-heptanedione (Hthd), and cerium (III) nitrate hexahydrate (Ce(NO_3_)_3_.6H_2_O) were purchased from Aldrich without further purification in this paper. General solvents including *n*-hexane and tetrahydrofuran (THF) were freshly distilled from sodium before use. 

### 3.2. Characterization

Structural measurement was performed by computer-controlled Oxford Xcalibur E diffractometer with graphite monochromated Cu-Ka radiation (*λ* = 1.54178 Å) at 150 (2) K. Data were corrected for absorption effects using the multi-scan technique (SADABS) [56]. The structure was solved by the direct method and refined by full-matrix least-squares methods on F^2^ using the SHELXTL program package [57]. H atoms attached to C atoms were treated as riding ones. All the non-hydrogen atoms were located from the Fourier maps and refined with anisotropic displacement parameters. The hydrogen atoms were positioned with idealized geometry and refined with fixed isotropic vibration parameters related to the non-H atom to which they were bonded. And the molecular structure was drawn by Diamond 3.2. 

The C, H, N analysis was examined using a Vario EL IIIElementar microanalyzer. Melting points were measured by the X-4A melting point apparatus. The film thickness was measured by an EOPTICS SE-100A spectroscopic ellipsometer (Wuhan, China), and the incident angle was fixed at 65°. The wavelength region from 350 to 1000 nm was scanned with a step of 1 nm. The thickness of the CeO_x_ film was 110 nm, which was measured by an ellipsometer and deposited on a SiO_2_/Si (100) wafer at 300 °C for 2200 cycles, and the result responded to the cross-sectional data by SEM. The ellipsometric thickness for CeO_x_ film prepared under the above conditions was averaged using at least three measurement points on every wafer. For the wafers prepared and measured in this study, the thickness at each of the different points did not exceed more than 2%. The growth rate was calculated from the film thickness and the number of deposition cycles. Bruker Multimode atomic force microscopy (AFM) was used to obtain the surface morphologies of the films. Thermo ESCALAB 250Xi photoelectron spectroscopy (XPS) was used to obtain the composition and crystalline phase of the film (Al Kα radiation for 1486.6 eV, survey step width for 1 eV, core level step width for 0.1 eV, calibrated with respect to the C 1s peak at 284.6 eV, put conductive tape under the sample, analysis pressure for 10^−10^ Pa, Ar+ kinetic energy: 1000 eV and sputtering for 2 min). The crystalline phase of the film was analyzed by X-ray diffraction (XRD) measurements (Rigaku Ultima IV, Cu-Ka radiation, X-ray wavelength: 1.54056 Å).

### 3.3. Synthesis 

#### 3.3.1. Synthesis of Ce(thd)_3_phen (**1**)

Based on the previous study, Ce(thd)_3_-phen (**1**) was synthesized as described in Figure 1 [49]. Complex **1** with a melting point of 208.1–209.8 °C was obtained as brown power (2.9 g, 85%) at −20 °C for 1 day. ^1^H-NMR (400 MHz, CDCl_3_, 25 °C, ppm) δ 12.08 (s, 3H, C=O***CH***C=O), 7.26 (s, 2H, (**3**)), 6.90 (s, 2H, (**4**)), 5.14 (s, 2H, (**5**)), 1.64 (s, 54H, ***^t^Bu***), −2.86 (s, 2H, H (**2**)).

#### 3.3.2. Synthesis of Ce(thd)_3_-MEDA (**2**)

The Ce(thd)_3_-MEDA (**2**) was synthesized in Figure 2. To a 100 mL Schlenk flask charged with NaOH (0.464 g, 11.6 mmol) and THF (20 mL), a mixed solution of Hthd (2.14 g, 11.6 mmol) and THF (5 mL) was then added dropwise over a 5 min period at 0 °C. Subsequently, the mixture was stirred at room temperature for 4 h. The resultant solution was then added dropwise to the solution Ce(NO_3_)_3_.6H_2_O (1.736 g, 4 mmol) in 20 mL of THF. A solution of MEDA (0.343 g, 4.4 mmol) in 5 mL of THF was added dropwise. The mixture was stirred for 8 h, and volatiles were removed in vacuo to obtain a yellow solid. Recrystallization from *n*-hexane gave complex **2** with a melting point of 157.3–161.3 °C as a pale yellow crystal at −20 °C for 2 days. Yield: 2.4 g (83%). ^1^H-NMR (400 MHz, C_6_D_6_, 25 °C, ppm) δ 11.15 (s, 3H, C=O***CH***C=O), 2.13 (s, 54H, ***tBu***), −1.97 (s, 3H, ***CH_3_***-NH-CH_2_-CH_2_-NH_2_), −2.55 (s, 2H, CH_3_-NH-***CH_2_***-CH_2_-NH_2_), −3.99 (s, 2H, CH_3_-NH-CH_2_-***CH_2_***-NH_2_), −12.03 (s, 1H, CH_3_-*NH*-CH_2_-CH_2_-NH_2_), −13.53 (s, 2H, CH_3_-NH-CH_2_-CH_2_-***NH_2_***). Calcd for C_36_H_65_CeN_2_O_6_: C, 56.7; H, 8.5; N, 3.7; Found: C, 56.5; H, 8.7; N, 4.0.

#### 3.3.3. Synthesis of Ce(thd)_3_-MOMA (**3**)

The Ce(thd)_3_-MOMA (**3**) was synthesized in Figure 3. Complex **3** was synthesized following the route used for complex **2** using NaOH (0.464 g, 11.6 mmol), Hthd (2.14 g, 11.6 mmol), Ce(NO_3_)_3_.6H_2_O (1.736 g, 4 mmol), and MOMA (0.392 g, 4.4 mmol). Recrystallization from *n*-hexane gave complex **3** with a melting point of 137.2–141.5 °C as a pale yellow crystal at −20 °C for 2 days. Yield: 2.5 g (80%). ^1^H-NMR (400 MHz, C_6_D_6_, 25 °C, ppm) δ 12.50 (s, 3H, C=O***CH***C=O), 2.34 (s, 54H, ***^t^Bu***), −3.83 (s, 2H, CH_3_-OH-CH_2_-C***H***_2_-NH-CH_3_), −5.44 (s, 3H, CH_3_-OH-CH_2_-CH_2_-NH-C***H***_3_), −6.65 (s, 2H, CH_3_-OH-C***H***_2_-CH_2_-NH-CH_3_), −7.21 (s, 3H, C***H***_3_-OH-CH_2_-CH_2_-NH-CH_3_), −13.32 (s, 1H, N***H***). Calcd for C_37_H_68_CeNO_7_: C, 56.9; H, 8.7; N, 1.8; Found: C, 57.0; H, 8.5; N, 2.0.

#### 3.3.4. Synthesis of Ce(thd)_3_-DMDE (**4**)

The Ce(thd)_3_-DMDE (**4**) was synthesized in Figure 4. Complex **4** was synthesized following the route used for complex **2** using NaOH (0.464 g, 11.6 mmol), Hthd (2.14 g, 11.6 mmol), Ce(NO_3_)_3_.6H_2_O (1.736 g, 4 mmol), and DMDE (0.392 g, 4.4 mmol). Recrystallization from *n*-hexane gave complex **4** with a melting point of 126.1–130.4 °C as a pale yellow crystal at −20 °C for 2 days. Yield: 2.7 g (85%). ^1^H-NMR (400 MHz, C_6_D_6_, 25 °C, ppm) δ 11.55 (s, 3H, C=O***CH***C=O), 2.09 (s, 54H, ***^t^Bu***), −2.71 (s, 6H, N(C***H_3_***)_2_-CH_2_-CH_2_-OH), −3.36 (s, 2H, N(CH_3_)_2_-***CH_2_***-CH_2_-OH), −5.57 (s, 2H, N(CH_3_)_2_-CH_2_-***CH_2_***-OH), −13.23 (s, 1H, N(CH_3_)_2_-CH_2_-CH_2_-O***H***). Calcd for C_37_H_67_CeNO_7_: C, 57.1; H, 8.6; N, 1.8; Found: C, 57.2; H, 8.5; N, 1.9.

### 3.4. Thermogravimetric Analysis

The stability of the complex was investigated using an STA 449 F3 analyzer in an argon atmosphere at a heating rate of 5 °C/min from 25 to 500 °C. Vapor pressures were estimated using a modified literature method, which used TG based on the Langmuir equation [58]:(1)$$P=\frac{dm}{dt}\sqrt{\frac{T}{M}}\frac{\sqrt{2\pi R}}{\alpha 1}$$
where P (Pa) was the vapor pressure at temperature T (K), dm/dt (Kg/s) was the rate of mass loss per unit surface area during the TG experiment; M (Kg/mol) was the molecular mass of the complex, R (J/mol*K) was the gas constant, and α_1_ was the vaporization coefficient. The Langmuir equation was rewritten as follows: P = kν, where $\nu =\frac{dm}{dT}\sqrt{\frac{T}{M}}$ was the material-dependent part of the Langmuir equation, and $k=\frac{\sqrt{2\pi R}}{\propto 1}$ depends on the thermogravimetric experiment parameters. A benzoic acid was used as a standard in the Antoine equation.
(2)$$\ln P=A-\frac{B}{T+C}$$

The temperature program was set to jump by increments of 5 °C, and then the derivative of mass with respect to temperature dm/dT was obtained from the liner regions of the isotherm steps for each temperature. These data were used to find the pressure P (Pa) as a function of T (°C).

### 3.5. ALD of CeO_2_ Film Details

To further prove that complexes were used as precursors of ALD, a commercial ALD reactor (MNT f-150–212r (Jiangsu MNT Micro and Nanotech Co., LTD., Wuxi, China) was used for deposition. The films were deposited on SiO_2_/Si (100) wafers, and the SiO_2_ thickness was measured to be 100 nm, which prevented the formation of silicates [59]. Complex **2** was used as an ALD precursor and kept at 170 °C to produce vapor 0.3 Torr/1 atm pressure. Ozone (O_3_) was obtained from oxygen gas (99.999%) in an ozone generator and used as an oxidizing agent. Under the flow of nitrogen (99.999%), the working pressure was maintained at 50–60 Pa, and nitrogen was also used as the pulse and purge gas. The organic matter of the SiO_2_/Si (100) wafer was cleaned in acetone and deionized water sequentially and then dried with a hair dryer before deposition.

## 4. Conclusions

In conclusion, to improve the thermal stability and volatility of heteroleptic Ce (III) precursors for ALD of Ce-containing thin films, we introduced small-molecule neutral ligands to satisfy charge neutrality and provide saturation of the metal coordination sphere. Moreover, the complexing ability of a nitrogen-containing bidentate ligand with a cerium ion was stronger than that of a mixed oxygen-nitrogen-containing bidentate ligand. Four heteroleptic Ce(III) complexes, complexed by a variety of neutral ligands, were synthesized and characterized by ^1^H-NMR, elemental analysis, and X-ray single-crystal diffraction. The thermogravimetric analysis and vapor pressure results showed that complex **2** was determined to be the most volatile, with a vapor pressure of 0.3 Torr at 170 °C, as a liquid precursor at that temperature. The potential of complex **2** as an ALD precursor was evaluated through an ALD reactor, and a CeO_2_ film on a SiO_2_/Si (100) wafer was successfully prepared. All the above analysis results demonstrated that complex **2** was capable of being an ALD precursor.

## Data Availability

The data presented in this study are available in this article.

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
