# Peer review of "Atomic Layer Deposition of CeO2 Film with a Novel Heteroleptic Ce(III) Complex"

_molecules, 2024, doi:10.3390/molecules29132987_

Round 1

Reviewer 1 Report

Comments and Suggestions for Authors

The authors reported four compounds, and detailed discussions and comparisons were conducted on their structural analysis and characterization. Further testing and characterization were conducted on the properties of some compounds. Suggest the author to make improvements and modifications based on the suggestions.

1 Why Complex 2 was selected as an ALD precursor to deposit a CeO2 film on a SiO2/Si (100) wafer? The selflimited deposition results demonstrated that complex 2 was a potential ALD precursor. How about the others? Has it also been tested, or is it determined based on the structure that there are no good features before deciding not to test?

2 When it comes to literature, I would like to emphasize that research articles should use the latest research results as references, and older reports should be appropriately referenced. This can reflect the progressiveness of research. However, in the current manuscript, the authors have cited a large number of earlier literatures, especially those from the period of ten to twenty years. This does not conform to the forefront of research and the characteristics of new discoveries. It is recommended that the author refer to similar literature, such as DOI10.1002/cjoc.202400076, these newer research reports and literature, etc. These literatures have new design strategies for structural design and are a good reference for current report.

3 The pixels in some images are not clear enough, as shown in Figure 6, etc. It is recommended to increase the pixels.

4 “The structure was solved using direct methods with SHELXS-97” in the experimental part, it should be corrected; now the SHELXL is updated, 2014 version is a normal one for the structure refinement.

Comments on the Quality of English Language

Appropriate language expression, some sentences need polishing.

Author Response

The authors reported four compounds, and detailed discussions and comparisons were conducted on their structural analysis and characterization. Further testing and characterization were conducted on the properties of some compounds. Suggest the author to make improvements and modifications based on the suggestions.

Comment 1: Why Complex 2 was selected as an ALD precursor to deposit a CeO2 film on a SiO2/Si (100) wafer? The selflimited deposition results demonstrated that complex 2 was a potential ALD precursor. How about the others? Has it also been tested, or is it determined based on the structure that there are no good features before deciding not to test?

Response 1: The TG results show that complexes 1 and 2 have single-step volatilization with no decomposition, which were potential ALD precursor. The thermal decomposition of complex 3 and 4 indicated that was not suitable for ALD precursor. The volatility and T50 results show that complex 2 has more excellent volatility than complex 1, and complex 2 was as a liquid precursor at sublimation temperature (170 oC was higher than the melting point). The complex 1 was as a solid precursor at sublimation temperature (170 oC was lower than the melting point).

Comment 2: When it comes to literature, I would like to emphasize that research articles should use the latest research results as references, and older reports should be appropriately referenced. This can reflect the progressiveness of research. However, in the current manuscript, the authors have cited a large number of earlier literatures, especially those from the period of ten to twenty years. This does not conform to the forefront of research and the characteristics of new discoveries. It is recommended that the author refer to similar literature, such as DOI10.1002/cjoc.202400076, these newer research reports and literature, etc. These literatures have new design strategies for structural design and are a good reference for current report.

Response 2: Thank you for your practical suggestion. The article was suitable for reference 45 in this paper.

Comment 3:The pixels in some images are not clear enough, as shown in Figure 6, etc. It is recommended to increase the pixels.

Answerer 3: Thank you for your practical suggestion. I have modified in Figure 6.

Comment 4: “The structure was solved using direct methods with SHELXS-97” in the experimental part, it should be corrected; now the SHELXL is updated, 2014 version is a normal one for the structure refinement.

Response 4: Thank you for your practical suggestion. I have modified in references 56 and 57. The reference 56 “Sheldrick, G.M. SHELXS-97 Program for Crystal Structure Solution. University of Göttingen Germany. 1997.” was corrected as “Sheldrick, G.M. Crystal Structure refinement with SHELXL. Acta .Crystallogr. C. 2015, 71, 3-8.”. The reference 57 “Sheldrick, G.M. SHELXL-97, Program for Crystal Structure Refinement. University of Göttingen Germany. 1997.” was corrected as “Sheldrick, G.M. SHELXT–Integrated space-group and crystal-structure determination, Acta. Crystallogr. A. 2015, 71, 3–8.”.

Reviewer 2 Report

Comments and Suggestions for Authors

The paper reports on the synthesis of four new heteroleptic cerium complexes containing dipivalylmethane ligand, their characterization and use as cerium oxide thin film precursors in atomic layer deposition process. The subject of the paper fits well the scope of Molecules journal.

I have the following comments:

1. Lines 21-24: Please specify, in which applications cerium dioxide could replace conventional silicon dioxide? The latter is not used in solid oxide fuel cells, catalysts, etc.

2. No information is provided on the method used for measuring cerium oxide film thickness. Please add the corresponding description into Section 3, Experimental.

3. Please provide a detailed description of the method used for determining vapor pressure of the complexes.

4. English should be improved throughout the manuscript.

Comments on the Quality of English Language

Moderate editing of English language required

Author Response

The paper reports on the synthesis of four new heteroleptic cerium complexes containing dipivalylmethane ligand, their characterization and use as cerium oxide thin film precursors in atomic layer deposition process. The subject of the paper fits well the scope of Molecules journal.

I have the following comments:

Comment 1: Lines 21-24: Please specify, in which applications cerium dioxide could replace conventional silicon dioxide? The latter is not used in solid oxide fuel cells, catalysts, etc.

Response 1: Thank you for your practical suggestion. I have modified in the paper accordingly. The statement of “these properties of CeO2 material can be used in applications such as solid oxide fuel cells, optical coatings, catalysts for water splitting and air purification” was corrected as “CeO2 material also can be used in applications such as solid oxide fuel cells, optical coatings, catalysts for water splitting and air purification”.

Comment 2: No information is provided on the method used for measuring cerium oxide film thickness. Please add the corresponding description into Section 3, Experimental.

Response 2: Thank you for your practical suggestion. I have modified in the paper accordingly. The statement of “The film thickness was measured by an EOPTICS SE-100A spectroscopic ellipsometer (Wuhan, China) using the Cauchy optical model and the results were calibrated by a Hitachi S-4800 scanning electron microscope (SEM).” was corrected as “The film thickness was measured by an EOPTICS SE-100A spectroscopic ellipsometer (Wuhan, China), and the incident angle was fixed at 65 o, the wavelength region from 350 to 1000 nm was scanned with a step of 1 nm. The thickness of the CeOx film was 110 nm, which was measured by an ellipsometer and deposited on SiO2/Si (100) wafer at 300 oC for 2200 cycles, and the result responded to the cross-sectional data by SEM. The ellipsometric thickness for CeOx film prepared under the above conditions was averaged using at least three measurement point on every wafer. For the wafers prepared and measured in this study, the thickness at each of the different points did not more than 2 %.”

Comment 3: Please provide a detailed description of the method used for determining vapor pressure of the complexes.

Response 3: Thank you for your practical suggestion. I have modified in the paper accordingly. The statement of “The vapor-temperature curve of the complex was obtained based on the Langmuir and Antoine equation using benzoic acid as a standard” was corrected as “Vapor pressures were estimated using a modified literature method, which using TG was based on the Langmuir equation.

 (1)

Where P was the vapor pressure at temperature T, dm/dt was the rate of mass loss per unit surface area during the TG experiment; M was the molecular mass of the complex, R was the gas constant and α1 was the vaporization coefficient. The Langmuir equation was rewritten as follows: P = kν, where  was the material-dependent part of the Langmuir equation, and  depends on the thermogravimetric experiment parameters, and using a benzoic acid as a standard in the Antoine equation.

 (2)

The temperature program was set to jump by increments of 5 oC, and then the derivative of mass respect to temperature dm/dT was obtained from the liner regions of the isotherm steps for each temperature. These data were used to find the pressure P as a function of T.”

Comment 4: English should be improved throughout the manuscript

Response 4: Thank you for your practical suggestion. I would try our best to improve the manuscript.
